# Neural Human Performer: Learning Generalizable Radiance Fields for Human Performance Rendering

**Youngjoong Kwon**[1], **Dahun Kim**[2], **Duygu Ceylan**[3], **Henry Fuchs**[1]
[1]University of North Carolina at Chapel Hill. [2]KAIST. [3]Adobe Research.
{youngjoong,fuchs}@cs.unc.edu   {mcahny}@kaist.ac.kr   {ceylan}@adobe.com

## Abstract

In this paper, we aim at synthesizing a free-viewpoint video of an arbitrary human performance using sparse multi-view cameras. Recently, several works have addressed this problem by learning person-specific neural radiance fields (NeRF) to capture the appearance of a particular human. In parallel, some work proposed to use pixel-aligned features to generalize radiance fields to arbitrary new scenes and objects. Adopting such generalization approaches to humans, however, is highly challenging due to the heavy occlusions and dynamic articulations of body parts. To tackle this, we propose Neural Human Performer, a novel approach that learns generalizable neural radiance fields based on a parametric human body model for robust performance capture. Specifically, we first introduce a temporal transformer that aggregates tracked visual features based on the skeletal body motion over time. Moreover, a multi-view transformer is proposed to perform cross-attention between the temporally-fused features and the pixel-aligned features at each time step to integrate observations on the fly from multiple views. Experiments on the ZJU-MoCap and AIST datasets show that our method significantly outperforms recent generalizable NeRF methods on unseen identities and poses. The video results and code are available at https://youngjoongunc.github.io/nhp.

## 1   Introduction

Free-viewpoint video of a human performer has a variety of applications in the area of telepresence, mixed reality, gaming and etc. Conventional free-viewpoint video systems require extremely expensive setups such as dense camera rigs [4, 7, 43] or accurate depth sensors [6, 13], to capture person-specific appearance information. In this paper, we aim at a scalable solution for free-viewpoint human performance rendering that generalizes across different human performers and requires only sparse camera views. However, representing and rendering arbitrary human performances is extremely challenging when the observations are highly sparse (up to three to four views) due to heavy self-occlusions and dynamic articulations of the body parts. In particular, an effective solution needs to coherently aggregate appearance information from sparse multi-view observations across time as the body undergoes a 3D motion. Furthermore, the solution needs to generalize to unseen motions and characters at test time.

Recently, neural radiance fields (NeRF) [26, 11, 17, 30, 33, 35, 36, 47, 50, 52, 53] have shown photo-realistic novel view synthesis results in per-scene optimization settings. To avoid the expensive per-scene training and improve the practicality, generalizable NeRFs [36, 52, 47] have been proposed which use image-conditioned, pixel-aligned features and achieve feed-forward view synthesis from sparse input views [36, 52]. Direct application of these methods to complex and non-rigid human motion is not straightforward, however, and naive solutions suffer from significant artifacts as shown in Fig. 3. Some existing methods [37, 52] aggregate image features across multiple views by simple average pooling, which often leads to over-smoothed outputs when details observed from multiple

35th Conference on Neural Information Processing Systems (NeurIPS 2021).

views (*e.g.*, front and side views) differ due to self occlusions of humans. Alternatively, several methods [23, 33] have proposed to learn person-specific global appearance features from multi-view observations. However, such methods are not able to generalize to new human performers.

To address these challenges, we propose Neural Human Performer, a novel approach that learns generalizable radiance fields based on a parametric 3D body model for robust performance capture. In addition to exploiting a parametric body model as a geometric prior, the core of our method is a combination of temporal and multi-view transformers which help to effectively aggregate spatio-temporal observations to robustly compute the density and color of a query point. First, the temporal transformer aggregates trackable visual features based on the input skeletal body motion across time. The following multi-view transformer performs cross-attention between the temporally-augmented skeletal features and the pixel-aligned features for each time step. The proposed modules collectively contribute to the adaptive aggregation of multi-time and multi-view information, resulting in significant improvements in synthesis results in different generalization settings of unseen motions and identities.

We study the efficacy of Neural Human Performer on two multi-view human performance capture datasets, ZJU-MoCap [33] and AIST [16]. Experiments show that our method significantly outperforms recent generalizable radience field (NeRF) methods. Furthermore, we compare ours with identity-specific methods [33, 44, 49] that also utilize a 3D human body model prior. Surprisingly, our generalized method achieves better rendering quality than the person-specific dedicated methods when tested on novel poses demonstrating the effectiveness of our transformer-based generalizable representation.

To summarize, our contributions are:

- We present a new feed-forward method of synthesizing novel-view videos of arbitrary human performers from sparse camera views. We propose Neural Human Performer that learns generalizable neural radiance representations by leveraging a 3D body motion prior.
- We design a combination of temporal and multi-view transformers that can aggregate information on the fly over video frames across multiple views to render each frame of the novel-view video.
- We show significant improvements over recent generalizable NeRF methods on unseen identities and poses. Moreover, our generalization results can outperform even person-specific methods when tested on unseen poses.

## 2   Related works

**Human performance capture.**   Novel view synthesis of human performance has a long history in computer vision and graphics. Traditional methods rely on complicated hardware such as dense camera rigs [4, 7, 43] or accurate depth sensors [6, 13]. To enable free-view video from sparse camera views, template-based methods [3, 5, 10, 42] exploit pre-scanned human models to track the motion of a person. However, their synthesis results are not photo-realistic and pre-scanned human models are not available in most cases. Recent methods [27, 37, 38, 54] learn 3D human geometry priors along with pixel aligned features to enable detailed 3D human reconstructions even from single images. However, these methods often suffer under complex human poses that are never seen during training and hence cannot be directly used for our purpose of human performance synthesis.

**Neural 3D representations.**   Recently, there has been great progress in learning neural networks to represent the shape and appearance of scenes. The 3D representations are learned from 2D images via differentiable rendering networks. Convolutional neural networks are used to predict volumetric representations via 3D voxel-grid features [40, 23, 29, 25, 14, 15], point clouds [1, 49], textured meshes [18, 21, 44] and multi-plane images [8, 55]. The learnt representations are projected by a 3D-to-2D operation to synthesize images. Some methods [14, 15] introduce consistency loss to improve the multi-view and temporal consistency. However, these methods often have difficulty in scaling to higher resolution due to memory restrictions.

To eschew these problems, implicit function-based methods [20, 22, 28, 41] learn a multi-layer perceptron that directly translates a 3D positional feature into a pixel generator. The more recent NeRF [26] learns implicit fields of density and color with a volume rendering technique and achieves photo-realistic view synthesis. Among many following NeRF extensions,  [30, 31, 35, 53, 17, 50]

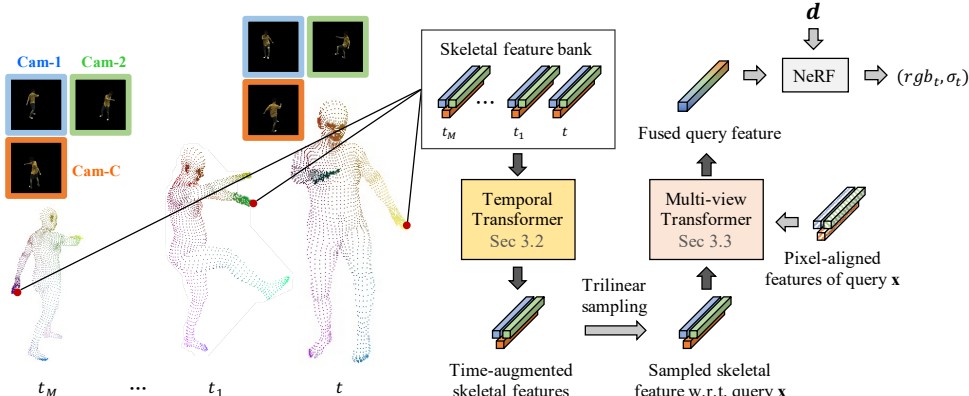

Figure 1: **Given a sparse set of multi-view videos of a person, Neural Human Performer uses temporal and multi-view transformers to learn pixel aligned features on the surface of a deformable body template such as SMPL. Such features are utilized to predict the color and density value of query points to render novel views of possibly unseen poses of the person.**

focus on dynamic scenes. While showing impressive results, it's an extremely under-constrained problem to jointly learn NeRF and highly dynamic deformation fields. To regularize the training, Neural Body [33] combines NeRF with a deformable human body model (*e.g.*, SMPL [24]). Despite the promising results, these general deformable NeRF [17, 53] and human-specific NeRF [11, 9, 33, 32] methods must be optimized for each new video separately, and generalize poorly on unseen scenarios. Generalizable NeRFs [36, 47, 52] try to avoid the expensive per-scene optimization by image-conditioning using pixel-aligned features. However, directly extending such methods to model complex and dynamic 3D humans is not straightforward when available observations are highly sparse. Unlike existing works, our method exploits temporal and multi-view information on-the-fly and achieves free-viewpoint human rendering in a *feed-forward* manner, also generalizing to new, unseen human identities and poses.

## 3 Method

### 3.1 Neural Human Performer

**Problem definition.** In our setting, given a sparse set (e.g., 3 or 4) of multi-view cameras $c = 1, \ldots, C$, input videos of an arbitrary human performance $I_{c,1:T} := \{I_{c,1}, I_{c,2}, ..., I_{c,T}\}$ are captured for each camera view $c$ defined by $\{\mathbf{K}_c, [\mathbf{R}|\mathbf{t}]_c\}$. We assume that the 3D human body model fit corresponding to each frame is given. Our goal is to synthesize a novel view video $\hat{I}_{q,1:T}$ for a query viewpoint $q$ defined by $\{\mathbf{K}_q, [\mathbf{R}|\mathbf{t}]_q\}$.

**Overview.** To compensate for the sparsity of available input views, we propose to exploit temporal information across video frames. In practice, we sample $M$ memory frames from the original input videos to augment each queried timestep $t$. Our goal is to learn generalizable 3D representations of human performers from multi-time ($M$) and multi-view ($C$) observations.

To this end, the Neural Human Performer is proposed with two main components as illustrated in Fig. 1. First, we construct the time-augmented skeletal features $\{s\}$. We exploit a human body model (SMPL [24]) to construct 3D skeletal features by projecting all the SMPL vertices onto each memory frame and picking up the pixel-aligned image features [36, 37, 52] at the projected 2D locations. The skeletal features are sampled from all memory frames to construct the skeletal feature bank. Inspired by Transformers[2, 46, 48], we propose a temporal Transformer that aggregates these memory features into the time-augmented skeletal features $\{s'\}$.

In the second stage, given a query 3D point $\mathbf{x}$, skeletal features are sampled at $\mathbf{x}$. In addition, pixel-aligned features $\{p\}$ at each time $t$ are sampled by directly projecting $\mathbf{x}$ onto the input images $\{I\}_t$. The multi-view Transformer is proposed to learn the correlation between the pixel-aligned features $\{p\}$ and the time-augmented skeletal features $\{s'\}$, and to adaptively fuse the multi-view

information. The final fused feature of the query point $\mathbf{x}$ is fed into the radiance field module to predict its color and density.

## 3.2 Construction of time-augmented skeletal features

Unlike static scenes, video inputs of moving characters inherently contain more visual cues as the occluded regions in a frame may be visible in other (potentially distant) frames. To take advantage of this temporal information, we first build up the skeletal feature bank from memory frames by leveraging a parametric body model (see Fig. 1). Then, we propose a temporal Transformer module that aggregates such collected features.

For each view $c$ and time $t$, we first build frame-level skeletal features $s_{c,t} \in \mathbb{R}^{L \times d}$ by sampling image features at the projection of the SMPL vertices in each image $I_{c,t} \in \mathbb{R}^{H \times W \times 3}$. $L$ denotes the number of SMPL vertices and $d$ is the dimension of image features.

After collecting all the skeletal features from all memory frames, we aggregate the time information in an attention-aware manner, instead of using simple average pooling. Specifically, for any $i^{th}$ skeletal feature vertex $s_{c,t}^i \in \mathbb{R}^d$, the proposed temporal Transformer casts attention over all other features contained in the vertex's memory bank $s_{c,t_1:t_M}^i = \{s_{c,t_1}^i, s_{c,t_2}^i, ..., s_{c,t_M}^i\} \in \mathbb{R}^{M \times d}$. In particular, we compute soft weights for all memory feature vertices in a non-local manner with respect to the current timestep $t$. Then, the values of the memory features are weighted summed as

$$
\begin{aligned}
t\_att_{c,t}^i &= \psi\big(\frac{1}{\sqrt{d_0}} q(s_{c,t}^i) \cdot k(s_{c,t_1:t_M}^i)^T\big), \quad t\_att_{c,t}^i \in \mathbb{R}^{1 \times M} \\
{s'_{c,t}}^i &= t\_att_{c,t}^i \cdot v(s_{c,t_1:t_M}^i) + s_{c,t}^i, \qquad s'_{c,t} \in \mathbb{R}^{L \times d} \quad \forall i
\end{aligned}
\tag{1}
$$

where $\psi$ represents the softmax operator along the second axis, $q(\cdot)$, $k(\cdot)$ and $v(\cdot)$ are learnable query, key and value embedding functions $\mathbb{R}^{d \to d_0}$ of the temporal Transformer.

In other words, the representation $s_{c,t}^i$ of each skeletal vertex at time $t$ is computed through a dynamically weighted combination of all its previous and next representations in the memory frames. This allows our network to incorporate helpful information and ignore irrelevant ones from other timesteps. In practice, the temporal Transformer operation in Eq. (1) is performed by a batch matrix multiplication for all skeletal vertices $L$ and all available viewpoints $C$.

## 3.3 Multi-view aggregation of skeletal and query features

Given a query 3D point $\mathbf{x} \in \mathbb{R}^3$, we retrieve the corresponding (time-augmented) skeletal feature ${s'_{c,t}}^{\mathbf{x}} \in \mathbb{R}^d$ at the queried location via trilinear interpolation in the SMPL space with SparseConvNet [19], following [39, 34, 51, 33].

In addition, we sample pixel-aligned image feature $p_{c,t}^{\mathbf{x}}$ via direct $\mathbb{R}^{3 \to 2}$ projection of the query point $\mathbf{x}$ on $I_{c,t}$. It is important to note that the pixel-aligned feature $p_{c,t}^{\mathbf{x}}$ is time-specific and represents the exact query location of $\mathbf{x}$, while the skeletal feature ${s'_{c,t}}^{\mathbf{x}}$ is time-augmented (w.r.t $t$) and contains inherent geometric deviations in the SMPL vertices and the following trilinear interpolations. We propose to combine these two complementary features, which will be shown to be effective in Sec. 4.4.

Given the two sets of multi-view features, skeletal ${s'_{1:C,t}}^{\mathbf{x}} = \{{s'_{c,t}}^{\mathbf{x}} | c = 1, ..., C\} \in \mathbb{R}^{C \times d}$ and pixel-aligned $p_{1:C,t}^{\mathbf{x}} = \{p_{c,t}^{\mathbf{x}} | c = 1, ..., C\} \in \mathbb{R}^{C \times d}$, we propose a multi-view Transformer that performs cross-attention from skeletal to pixel-aligned features. Specifically, the values of pixel-aligned features from all viewpoints is re-weighted based on how much compatible they are with each skeletal features. The non-local cross-attention $mv\_att$ is constructed as:

$$
\begin{aligned}
mv\_att_t^{\mathbf{x}} &= \psi\big(\frac{1}{\sqrt{d_1}} k({s'_{1:C,t}}^{\mathbf{x}}) \cdot k(p_{1:C,t}^{\mathbf{x}})^T\big), \quad mv\_att_t^{\mathbf{x}} \in \mathbb{R}^{C \times C} \\
z_{1:C,t}^{\mathbf{x}} &= mv\_att_t^{\mathbf{x}} \cdot v(p_{1:C,t}^{\mathbf{x}}) + v({s'_{1:C,t}}^{\mathbf{x}}), \qquad z_{1:C,t}^{\mathbf{x}} \in \mathbb{R}^{C \times d}, \quad z_{c,t}^{\mathbf{x}} \in \mathbb{R}^{1 \times d}
\end{aligned}
\tag{2}
$$

where $\psi$ represents the softmax operator along the second axis. Note that $k$ and $v$ are new layers independent from those in the temporal Transformer. The confident observations in each view will have large weights and be highlighted, and vice versa. Finally, we use the view-wise mean of $z_t^{\mathbf{x}} = \frac{1}{C} \sum_c z_{c,t}^{\mathbf{x}} \in \mathbb{R}^d$ as our *meta-time* and *meta-view* representation of the query point $x$.

The final density $\sigma_t(\mathbf{x})$ and color values $rgb_t(\mathbf{x})$ at time $t$ are computed as:

$$\sigma_t(\mathbf{x}) = MLP_\sigma(z_t^\mathbf{x}), \qquad rgb_t(\mathbf{X}) = MLP_\mathbf{rgb}(\sum_c (z_{c,t}^\mathbf{x}; \gamma_\mathbf{d}(\mathbf{d}))/C), \qquad (3)$$

where $MLP_\sigma$ and $MLP_\mathbf{rgb}$ consist of four and two linear layers respectively, and $\gamma_\mathbf{d} : \mathbb{R}^{3 \rightarrow 6 \times l}$ is a positional encoding of viewing direction $\mathbf{d} \in \mathbb{R}^3$ as in [26] with $2 \times l$ different basis functions.

More details on the network architecture can be found in the supplementary material.

### 3.4 Volume Rendering

The predicted color of a pixel $p \in \mathbb{R}^2$ for a target viewpoint $q$ in the focal plane of the camera and center $\mathbf{r}_0 \in \mathbb{R}^3$ is obtained by marching rays into the scene using the camera-to-world projection matrix, $\mathbf{P}^{-1} = [\mathbf{R}_q|\mathbf{t}_q]^{-1}\mathbf{K}_q^{-1}$ with the direction of the rays given by $\mathbf{d} = \frac{\mathbf{P}^{-1}p - \mathbf{r}_0}{\|\mathbf{P}^{-1}p - \mathbf{r}_0\|}$.

We then accumulate the radiance and opacity along the ray $\mathbf{r}(z) = \mathbf{r}_0 + z\mathbf{d}$ for $z \in [z_\text{near}, z_\text{far}]$ as defined in NeRF [26] as follows:

$$\mathbf{I}_q(p) = \int_{z_\text{near}}^{z_\text{far}} \mathbf{T}(z)\sigma(\mathbf{r}(z))\mathbf{c}(\mathbf{r}(z), \mathbf{d})dz, \quad where \quad \mathbf{T}(z) = \exp\left(-\int_{z_\text{near}}^z \sigma(\mathbf{r}(s))ds\right) \qquad (4)$$

In practice, we uniformly sample a set of 64 points $z \sim [z_{near}, z_{far}]$. We set $\mathbf{X} = \mathbf{r}(z)$ and use the quadrature rule to approximate the integral. We compute the 3D bounding box of the SMPL parameters at time $t$ and derive the bounds for ray sampling $z_{near}, z_{far}$.

### 3.5 Loss Function

Given the ground truth target images $\mathbf{I}_{q,t}$, we train both the radiance field and feature extraction networks using a simple photometric reconstruction loss $\mathcal{L} = \|\hat{\mathbf{I}}_{q,t} - \mathbf{I}_{q,t}\|_2$ .

## 4 Experiments

We present novel view synthesis and 3d reconstruction results of human performances in different generalization scenarios. We compare our method against the current best view-synthesis methods from two classes: per-subject optimization methods that also use a human body model prior (Sec. 4.1) and generalizable NeRF methods (Sec. 4.2). We experiment on the ZJU-MoCap [33] and AIST datasets [45, 16]. For training and testing our model as well as the baselines, we remove the background using the foreground masks that are either provided by the dataset or pre-computed using an off-the-shelf method. Unless otherwise specified, we sample two memory frames $\{t - 20, t + 20\}$ at time $t$ (i.e., three timesteps in total) and take three canonical input views in all experiments. The details of the implementation, datasets, training process are provided in the supplementary material.

### 4.1 Comparison with body model-based, per-subject optimization methods.

**Baselines.** We perform comparisons with the state-of-the-art Neural Body (NB) [33] that combines the body model prior SMPL and NeRF in a per-subject optimization setting. Neural Textures (NT) [44] renders a coarse mesh with latent texture maps and uses a 2D CNN to render target images. We use the SMPL mesh as the coarse mesh input to NT in our experiments. NHR [49] extracts 3D features from input point clouds and renders them into 2D images. Since dense point clouds are difficult to obtain from sparse camera views, we use the SMPL vertices as input point clouds. These methods have reported that they can adapt to new poses of the same performer, hence we compare to them for the task of novel pose synthesis.

**Setup.** We experiment with ZJU-MoCap dataset [33] which provides performance captures of 10 human subjects captured from 23 synchronized cameras, human body model parameters as well as the foreground mask corresponding to each frame. Each video contains complex motions such as kicking and Taichi and is between 1000 to 2000 frames long. We first split the dataset into two parts: source and target videos. In all comparisons, the first 300 frames of either source or target videos are used during training, and the remaining next frames (unseen poses) are held out for testing. Note that

| Method | PSNR | SSIM |
|---|---|---|
| Trained on **source** subjects | | |
| NB | 23.79 | 0.887 |
| NHR | 22.31 | 0.871 |
| NT | 22.28 | 0.872 |
| Trained on **source** subjects | | |
| Ours | **26.94** | **0.929** |

a . Test results on **unseen** poses of **source** subjects

| Method | PSNR | SSIM |
|---|---|---|
| Trained on **target** subjects | | |
| NB | 22.88 | 0.880 |
| NHR | 22.03 | 0.875 |
| NT | 21.92 | 0.873 |
| Trained on **source** subjects | | |
| Ours | **24.75** | **0.906** |

b . Test results on **unseen** poses of **target** subjects

| Method | | PSNR | SSIM |
|---|---|---|---|
| Trained on **source** subjects | | | |
| NB | | 28.51 | **0.947** |
| NHR | | 23.95 | 0.897 |
| NT | | 23.86 | 0.896 |
| Trained on **source** subjects | | | |
| Ours | | **28.73** | 0.936 |
| Ours | *per-subject* | **31.57** | **0.966** |

c . Test results on **seen** poses of **source** subjects

Table 1: **Comparison with other body model-based, per-subject optimization methods.**

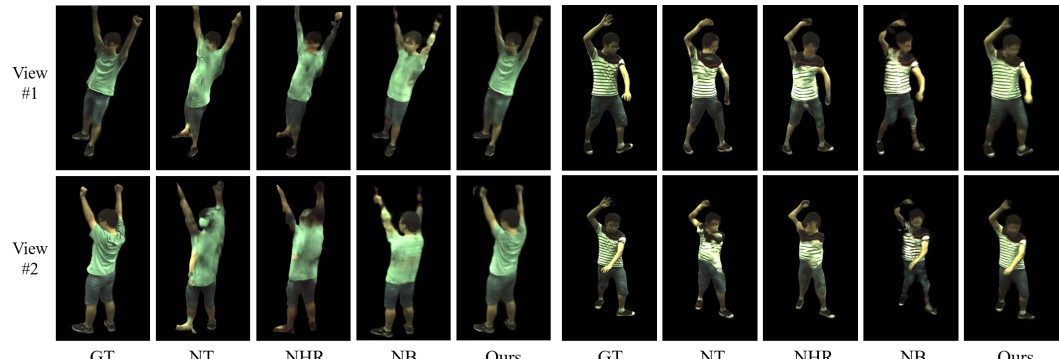

Figure 2: **Pose generalization – comparison with per-scene optimization methods that use a human body prior.** Results of NT(Neural textures) [44], NHR (Neural human rendering) [49], NB (Neural body) [33] and ours. Novel view synthesis on the ZJU-MoCap for **unseen** poses of **source** subjects. While ours is a single model trained on all **source** subjects; baselines are trained in a per-subject manner.

the baseline methods are always trained in a per-subject manner. To validate whether the training is reproducible, we experiment with 5 independent runs with random train/test splits and observe a variance of 0.15 PSNR, showing that the results are consistent. In each independent run, we used seven models for training and the other three for testing.

**Results.** We present three different comparison settings to validate our method. We would like to point out that all the comparison settings place our method ('ours') in disadvantage. This is because our model is trained on all the source subjects at once (one network for all subjects), while the competing methods are per-subject trained on the *subject to be tested* (one network for one subject) - easier setting. First, we evaluate the **pose generalization** capability (Table. 1a and Fig. 2) where we train on the training splits of the source subjects and test on the testing splits, i.e., unseen poses, of the same source subjects. Our method significantly outperforms all the baselines and the state-of-the-art Neural Body [33] by **+3.15** PSNR and +0.042 SSIM scores. We next evaluate the **identity generalization** capability (Table. 1b) by testing on the unseen poses of target subjects. While our method is trained on source subjects only, baselines are trained on target subjects. Note that this comparison is *disadvantageous* to us since unlike our method the baselines have *seen* the testing subjects as they must be trained separately for each subject. Surprisingly, however, our method still outperforms all the baselines by a healthy margin of +1.87 PSNR and +0.026 SSIM scores. These results indicate that our proposed architecture with the temporal and the multi-view transformers generalizes well across unseen identities and poses, and produces photo-realistic results. Finally, in Table. 1c, we evaluate our method and baselines on **seen poses of seen subjects**. Even when we train a single model of our method for all the source sobjects ('ours'), ours achieves comparable results with the state-of-the-art per-subject based methods [33]. To make a more fair comparison, we also train our method in the same per-subject setting similar to the baselines. Table. 1c shows that in this case, our model ('ours per-subject') achieves a significant improvement over the best-performing baseline [33] by +3 PSNR and +1.4% SSIM.

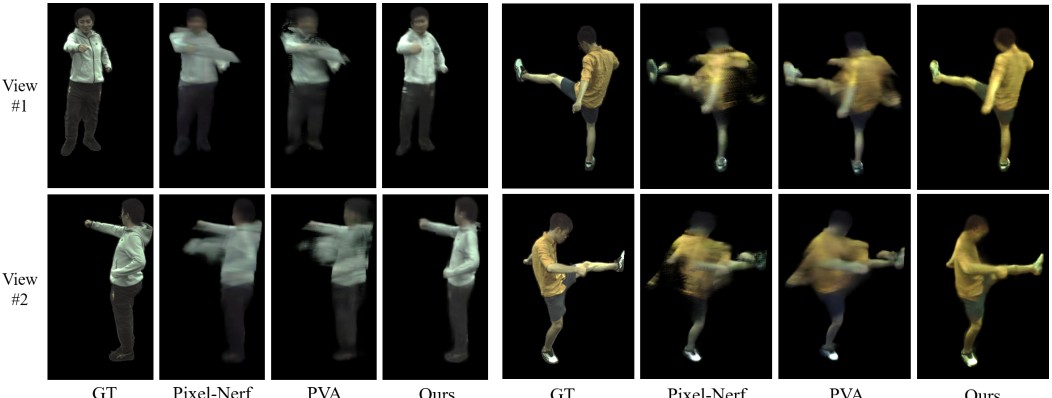

Figure 3: **Identity-and-pose generalization – comparison with generalizable NeRF methods.** Results of Pixel-Nerf [52], PVA (Pixel-aligned volumetric avatar) [36] and ours. Novel view synthesis on the ZJU-MoCap dataset where all methods are trained on **source** subjects and tested on **unseen** poses of **target** subjects.

| Method | PSNR | SSIM |
|---|---|---|
| Pixel-NeRF | 23.17 | 0.8693 |
| PVA | 23.15 | 0.8663 |
| Ours | **24.75** | **0.9058** |

a . Generalization results on ZJU-MoCap.

| Method | PSNR | SSIM |
|---|---|---|
| Pixel-NeRF | 18.06 | 0.7304 |
| PVA | 17.82 | 0.7211 |
| Ours | **19.03** | **0.8390** |

b . Generalization results on AIST.

Table 2: **Comparison with generalizable NeRF methods.**

## 4.2 Comparison with generalizable NeRF methods.

**Baselines.** Among the recent generalizable NeRF methods [36, 52, 47], we compare with Pixel-NeRF [52] and PVA [36] which focus on very sparse (up to 3 or 4) input views. We reimplement [36] since the code is not available.

**Setup.** In addition to the ZJU-MoCap dataset (see Sec. 4.1), we experiment on the larger AIST dataset [45, 16] to further evaluate the generalization abilities of different methods. AIST dataset provides dance videos of 30 human subjects captured from 9 cameras, together with the corresponding SMPL fits. We extract the foregroud mask of each image using an off-the-shelf human parser [12]. AIST dataset contains highly diverse motions, slow to fast, simple to complex. We split the dataset into 20 and 10 subjects for training and testing respectively, where the testing dataset contains unseen poses of unseen subjects.

**Novel view synthesis results.** As shown in Table. 2, across all the datasets and all the metrics, our method consistently outperforms the baselines by healthy margins of +1.6 PSNR and +0.037 SSIM scores. Fig. 3 and Fig. 5 present the same tendency in visualizations. Pixel-NeRF and PVA aggregate multi-view observations via average pooling without explicitly considering the correlation between the views. In contrast, our temporal and multi-view transformers learn to model the correlation between input views and integrate different observations to help the NeRF module to produce higher quality results. Another advantage of our method is that the used body model prior provides a robust geometric cue to handle the self-occlusion of the human body.

**3D reconstruction results.** We also evaluate the 3D reconstruction quality of generalizable NeRF methods and our method on the ZJU-MoCap (Fig. 4) and AIST datasets (Fig. 5) given three input views. The visualizations show that our 3D reconstructions align well with the input images, and is more accurate than even the per-subject method [33] (*e.g.*, the shape of the upper cloth in Fig. 4).

Overall, these results verify that since humans are complex and prone to occlusions, more sophisticated designs compared to simple image-conditioning are required to learn robust and accurate appearance representations.

## 4.3 Cross-dataset generalization.

We further study the generalization ability of our method across different datasets by training on one dataset and testing on another one as shown in Table. 4a. We would like to point out that the two

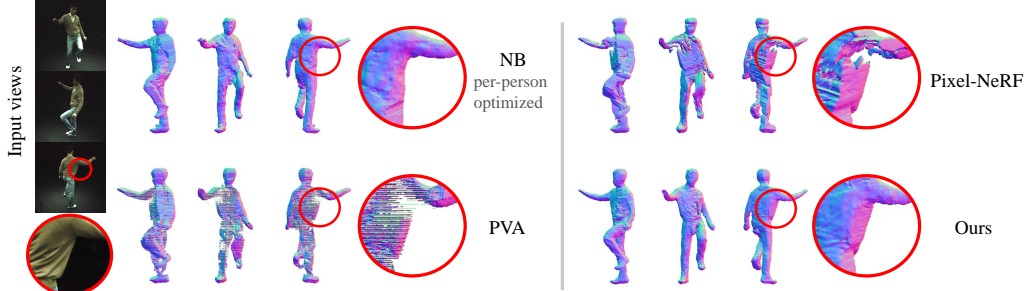

Figure 4: **3D reconstruction on ZJU-MoCap.** Tested on unseen poses of unseen subjects (except Neural Body which is a subject specific method). NB (Neural Body) [33], PVA (Pixel volumetric avatar) [36], Pixel-NeRF [52] and ours.

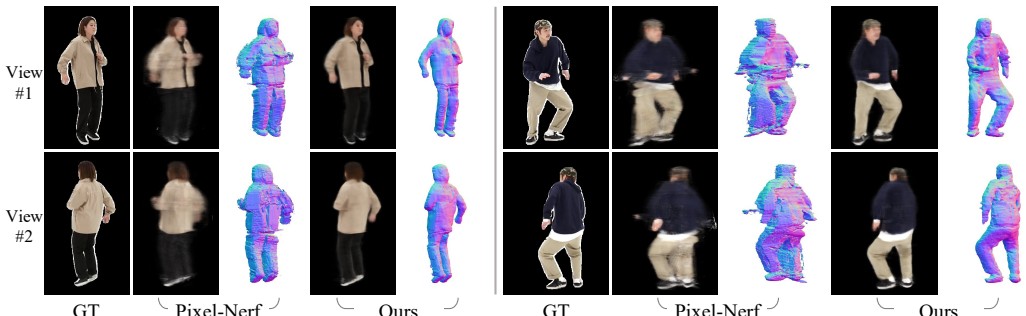

Figure 5: **Generalization results on AIST.** Novel view synthesis and 3D reconstruction results on unseen poses of unseen subjects.

datasets [33, 16] have significantly different statistics both in terms of color distribution (background, lighting) and distance of the camera to the subject, making the cross-dataset generalization task extremely challenging. Nevertheless, we found that only 8-minute fine-tuning on the target dataset can already outperform the baselines fully-trained on the target dataset. 16-minute fine-tuning performs on par with our model fully-trained on the target dataset.

## 4.4 Ablation studies

Table. 3 shows the ablation study on the ZJU-MoCap dataset for unseen subjects and unseen poses, using three time-steps and three camera views as input. Note that all the variations without either temporal or multi-view transformer modules use simple average pooling instead, to fuse temporal or multi-view observations respectively.

**Complementariness of skeletal and pixel-aligned query features.** 'Sk' uses only time-augmented skeletal features (Sec. 3.2) without time-specific pixel-aligned features, while 'Px' uses only the time-specific pixel-aligned features, on the contrary. Both 'only' models show the largest drops compared to our full model, and 'Sk + Px' model improves them by +1.2 PSNR and +0.9 PSNR respectively. This validates the complementariness between the skeletal and pixel-aligned features in that one is time-augmented but involves slight geometric deviations, while another is time-specific and represents exact query location, as discussed in Sec. 3.3.

**Impact of temporal and multi-view transformers.** 'Sk + Px' uses no transformers so far, falling behind our full model by -1.3 PSNR score. Then 'Sk + Px + T' adds the temporal transformer and improves +0.7 PSNR score, showing its effectiveness in aggregating information over video frames. 'Sk + Px + MV' uses the multi-view transformer module and shows the largest gain of +1.0 PSNR, indicating the efficacy of learnt cross-attention between the skeletal features and pixel-aligned features, as well as the importance of learnt inter-view correlations. Our full model 'Sk + Px + T + MV' shows the best use of all the proposed components and achieves 24.75 PSNR and 0.9058 SSIM.

**Impact of number of camera views.** Table. 4b shows the performance of our method when tested with different number of input views, fixing the number of timesteps to one. As expected, the performance degrades slightly with fewer input views (e.g., as few as one).

| Variant | Skeletal | Pixel-aligned | T-transformer | MV-transformer | PSNR | SSIM |
|---|---|---|---|---|---|---|
| Sk | ✓ | | | | 22.31 | 0.8865 |
| Px | | ✓ | | | 22.58 | 0.8780 |
| Sk + Px | ✓ | ✓ | | | 23.47 | 0.8906 |
| Sk + Px + T | ✓ | ✓ | ✓ | | 24.21 | 0.9016 |
| Sk + Px + MV | ✓ | ✓ | | ✓ | 24.44 | 0.9034 |
| Sk + Px + T + MV | ✓ | ✓ | ✓ | ✓ | **24.75** | **0.9058** |

Table 3: **Ablation study.** Results on the ZJU-MoCap dataset for unseen subjects and unseen poses. Sk: skeletal features, Px: pixel-aligned features, T: temporal transformer, MV: multi-view transformer.

| Exp. protocol | Fine-tune | PSNR | SSIM |
|---|---|---|---|
| Trained on AIST | 8-min | 24.25 | 0.8954 |
| Fine-tuned on ZJU-Mocap | 16-min | 24.73 | 0.9023 |
| Trained on ZJU-Mocap | 8-min | 18.63 | 0.8242 |
| Fine-tuned on AIST | 16-min | 18.83 | 0.8374 |

a . Cross-dataset generalization.

| # Timesteps | # Views | PSNR | SSIM |
|---|---|---|---|
| 1 | 1 | 20.13 | 0.835 |
| | 2 | 21.82 | 0.871 |
| | 3 | 23.33 | 0.906 |
| | 4 | 23.51 | 0.913 |

b . Impact of number of camera views.

Table 4: Cross-dataset generalization (left) and impact of different number of camera views tested on the ZJU-Mocap dataset for unseen poses of unseen subjects (right).

# 5 Limitations

While our novel generalizable human performance rendering method outperforms recent per-subject and generalizable NERF methods, there are remaining challenges yet to be explored. 1) While we show that our method can generalize across datasets with finetuning, the generalization capability will be limited as the distribution of the datasets become significantly different. 2) The performance of our method will be affected as the SMPL parameter accuracy degrades. This is the reason for lower performance on the AIST dataset where the accuracy of the SMPL fits are low due to highly complex motion sequences. It is an interesting direction to jointly refine the SMPL parameters using differentiable rendering within our framework. 3) Our algorithm does not have an explicit assumption of static cameras. However, in practice, it might be hard to estimate the inputs to our method (SMPL fits, camera parameters, foreground masks) with moving cameras due to motion blur, changing background, lighting, and synchronization issues etc. We consider this as an orthogonal problem and expect that any advancements in unconstrained multi-view capture setups will help to generalize our method to in-the-wild settings.

# 6 Societal impact

We discuss the potential societal impact of our work. On a positive note, the human performance synthesis is the key component of realizing telepresence, which has become more important especially in this pandemic era. In the future, people physically apart can feel like they are in the same space and feel connected with a few inexpensive webcams and AR/VR headsets thanks to the development of the telepresence. Like any other technology, our method can also be used for other reasons by corporations, e.g., for identity recognition from a small number of surveillance cameras. We strongly hope that our research will be used in positive directions.

# 7 Conclusion

We present Neural Human Performer, a generalizable radiance field network based on a parametric body model that can synthesize free-viewpoint videos for arbitrary human performers from sparse camera views. Leveraging the trackable visual features from the input body motion prior, we propose a combination of a temporal and a multi-view Transformer that integrates multi-time and multi-view observations in a feed-forward manner. Our method can produce photo-realistic view synthesis of unseen poses and identities at test time. In various generalization settings on the ZJU-MoCap and AIST datasets, our method achieves state-of-the-art performance compared to other per-subject methods that utilize a human body prior as well as the generalizable NeRF methods.

## Acknowledgments and Disclosure of Funding

We thank Sida Peng of Zhejiang University, Hangzhou, China, for many very helpful discussions on a variety of implementation details of the Neural Body [33]. We thank Ruilong li and Alex Yu of UC Berkeley for many discussions on the AIST++ dataset [16] and Pixel-NeRF [52] details. We thank Prof. Alex Berg of UNC for the generous offer of computational resources and Misha Shvets of UNC for a useful tutorial on it. This work was partially supported by National Science Foundation Award 1840131.

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
