# Neural Human Performer: Learning Generalizable Radiance Fields for Human Performance Rendering

**Youngjoong Kwon**[1], **Dahun Kim**[2], **Duygu Ceylan**[3], **Henry Fuchs**[1]
[1]University of North Carolina at Chapel Hill. [2]KAIST. [3]Adobe Research.
{youngjoong,fuchs}@cs.unc.edu  {mcahny}@kaist.ac.kr  {ceylan}@adobe.com

## A  Video results

Video results on the free-viewpoint rendering and 3D reconstruction with the ZJU-MoCap and AIST datasets can be found at https://youngjoongunc.github.io/nhp.

## B  Reproducibility

### B.1  Implementation details.

We describe the implementation details in the interest of reproducibility. Note that due to the high computing cost, we did not spend significant effort to tune the architecture or training procedure, and it is possible that variations can perform better, or that smaller models may suffice. Code will be made public upon publication.

**Image feature extractor.**    As briefly discussed in the main paper, we use an ImageNet-pretrained ResNet18 backbone to extract a feature pyramid. For an image of shape H×W, we take the multi-scale feature maps of shapes

$$\{ \ 64 \times \text{H/2} \times \text{W/2}, \quad 64 \times \text{H/4} \times \text{W/4}, \quad 128 \times \text{H/8} \times \text{W/8}, \quad 256 \times \text{H/16} \times \text{W/16} \ \}.$$

These feature maps are bilinearly upsampled to the highest resolution *i.e.*, H/2 × W/2, and concatenated into a shape 512 × H/2 × W/2.

**Temporal transformer.**    The temporal Transformer is used in construction of time-augmented skeletal features in Section 3.2 of the manuscript. The overview of the attention between the skeletal feature at $t$ and skeletal *memory* features is illustrated in Fig. 1. All the $L$ vertices are processed batch-wise, where the attention is computed for each vertex. $d$ is set to 64.

**Sampling of time-augmented skeletal feature w.r.t. a query point x.**    When we are given a query point $\mathbf{x}$ in 3D space, we sample the corresponding feature at $\mathbf{x}$'s 3D location, $s'^{\mathbf{x}}_{1:C,t} \in \mathbb{R}^{C \times d}$, from the previously constructed time-augmented skeletal features $s'_{1:C,t} \in \mathbb{R}^{L \times C \times d}$. Inspired by [11, 9, 8, 7], we adopt the SparseConvNet [6] to perform such sampling, whose architecture is described in Table 1. First, we compute the 3D bounding box of the human body based on the SMPL parameters, and divide the 3D box into small voxels of size of $5mm \times 5mm \times 5mm$, resulting in a $D' \times H' \times W'$ (depth, height, width) volume. The SparseConvNet consists in 3D sparse convolutions to process the input volume, diffusing the skeletal features into the nearby 3D space. We resize and concatenate the multi-scale outputs from the 5, 9, 13, 17-th layers as the output feature $\in \mathbb{R}^{\frac{D'}{16} \times \frac{H'}{16} \times \frac{W'}{16} \times 384}$. Since the diffusion of the skeletal feature should not be affected by the human position and orientation in the world coordinate system, we transform the skeletal feature locations to the SMPL coordinate system. Then, the query location $\mathbf{x}$ is also transformed to the SMPL coordinate system, and the corresponding skeletal feature $s'^{\mathbf{x}}_{1:C,t} \in \mathbb{R}^{C \times 384}$ is sampled via trilinear

35th Conference on Neural Information Processing Systems (NeurIPS 2021).

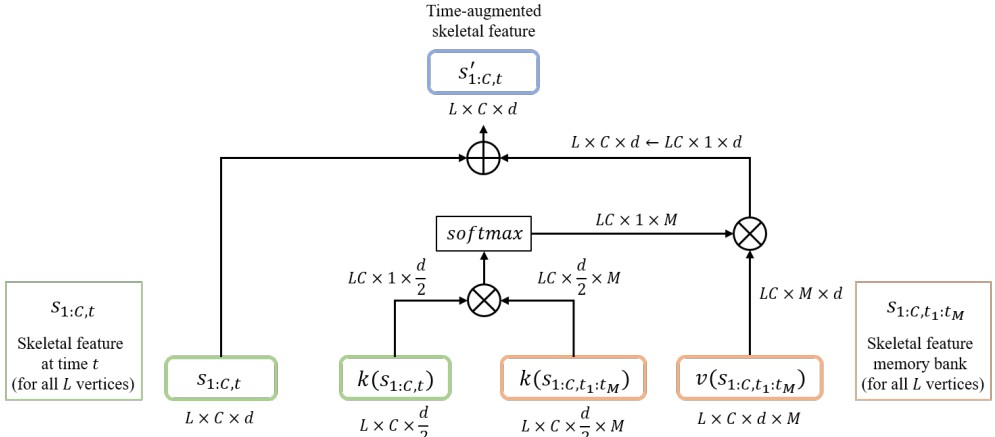

Figure 1: **Overview of temporal Transformer's attention between the skeletal features at $t$ and skeletal *memory* features.**

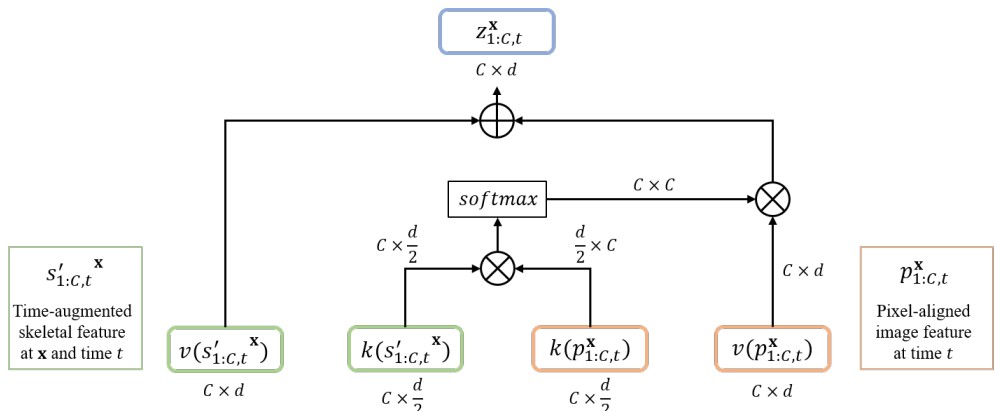

Figure 2: **Overview of multi-view Transformer's cross-attention between the sampled time-augmented skeletal features and time-specific pixel-aligned feature at $t$.**

interpolation, and a fully-connected layer reduces the channel-size to 128. The resulting skeletal feature $s'_{1:C,t}{}^{\mathbf{x}} \in \mathbb{R}^{C \times 128}$ is fed into the following multi-view transformer.

**Multi-view transformer.** The sampled time-augmented skeletal feature $s'_{1:C,t}{}^{\mathbf{x}}$ is fed into the proposed multi-view transformer to obtain our *meta-time* and *meta-view* representation of the query point $\mathbf{x}$, which is explained in Section 3.3 of the manuscript. The overview of the cross-attention between the sampled time-augmented skeletal features and time-specific pixel-aligned features is illustrated in Fig. 2. $d$ is set as 128.

**NeRF network.** The NeRF network takes the final representation from above $z^{\mathbf{x}}_{1:C,t}$ as input and predicts density $\sigma^{\mathbf{x}}_t$ and color $rgb^{\mathbf{x}}_t$. It consists of the fully-connected layers as illustrated in Fig. 3.

**Query point sampling details.** We first compute the 3D bounding box of the human subject from the corresponding SMPL vertice coordinates. Since there is a gap between the exact human subject geometry and the SMPL model, we enlarge the side length of the bounding box by 2.5% and this becomes the query point sampling bounds. We sample 1024 rays, and 64 points are sampled per ray for the training. For the inference, 64 points are sampled along each ray.

| | Layer Description | Output Dim. |
|---|---|---|
| | Input volume | D' $\times$ H' $\times$ W' $\times$ 64 |
| 1-2 | (3 $\times$ 3 $\times$ 3 conv, 64 features, stride 1) $\times$ 2 | D' $\times$ H' $\times$ W' $\times$ 64 |
| 3 | (3 $\times$ 3 $\times$ 3 conv, 64 features, stride 2) | D'/2 $\times$ H'/2 $\times$ W'/2 $\times$ 64 |
| 4-5 | (3 $\times$ 3 $\times$ 3 conv, 64 features, stride 1) $\times$ 2 | D'/2 $\times$ H'/2 $\times$ W'/2 $\times$ 64 |
| 6 | (3 $\times$ 3 $\times$ 3 conv, 64 features, stride 2) | D'/4 $\times$ H'/4 $\times$ W'/4 $\times$ 64 |
| 7-9 | (3 $\times$ 3 $\times$ 3 conv, 64 features, stride 1) $\times$ 3 | D'/4 $\times$ H'/4 $\times$ W'/4 $\times$ 128 |
| 10 | (3 $\times$ 3 $\times$ 3 conv, 128 features, stride 2) | D'/8 $\times$ H'/8 $\times$ W'/8 $\times$ 128 |
| 11-13 | (3 $\times$ 3 $\times$ 3 conv, 128 features, stride 1) $\times$ 3 | D'/8 $\times$ H'/8 $\times$ W'/8 $\times$ 128 |
| 14 | (3 $\times$ 3 $\times$ 3 conv, 128 features, stride 2) | D'/16 $\times$ H'/16 $\times$ W'/16 $\times$ 128 |
| 15-17 | (3 $\times$ 3 $\times$ 3 conv, 128 features, stride 1) $\times$ 3 | D'/16 $\times$ H'/16 $\times$ W'/16 $\times$ 128 |
| | Resize & Concat. outputs of layer 5, 9, 13, and 17 | D'/16 $\times$ H'/16 $\times$ W'/16 $\times$ 384 |

Table 1: **Architecture of SparseConvNet.** Each layer consists of sparse convolution, batch normalization and ReLU.

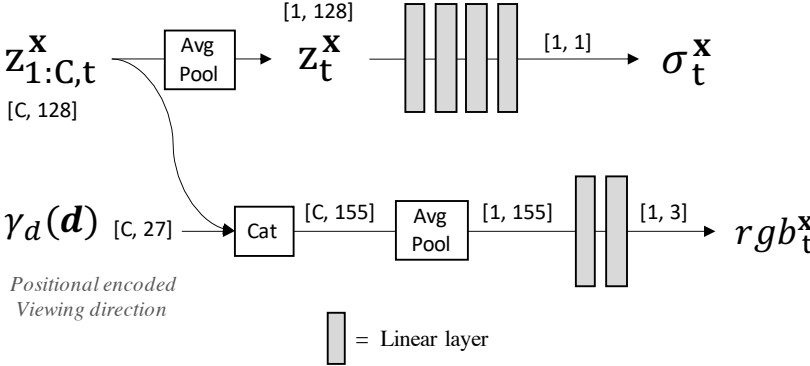

Figure 3: **Overview of NeRF architecture.**

## C  Datasets

We discuss the additional details about the datasets used, including the train/test splits and license information. Note that both the ZJU-Mocap and AIST datasets do not contain any personally identifiable information or offensive content.

### C.1  ZJU-MoCap

We use the $512 \times 512$ videos for the training and testing following the original Neural Body [7]. ZJU-Mocap provides 10 human subjects, and we reserved 7 for the training and 3 for testing on unseen identities. As mentioned in the main paper, we experiment with 5 independent runs with random train/test splits. For the qualitative results, we used subject 387, 393, 394 for the testing. ZJU-Mocap provides SMPL parameters obtained using EasyMocap[1] [2, 7, 3, 1] and foreground mask extracted using PGN [4]. ZJU-Mocap is the public dataset that is only meant for the research purposes as stated in their GitHub page.

### C.2  AIST

The original AIST dataset provides 60 fps videos with $1080 \times 1920$ resolutions [10] with corresponding SMPL parameters [5] obtained using AIST++ API[2]. AIST dataset does not provide foreground mask, so we obtained the foreground mask using PGN [4]. Since most part of the images are background, we center-crop the video to $800 \times 800$ sizes. During the training and evaluation, we resize the center-cropped video to $512 \times 512$. AIST contains 30 human subjects. We split the train and testing sets based on different subjects, which also makes sure the human motions in the train (20

---

[1]https://github.com/zju3dv/EasyMocap
[2]https://github.com/google/aistplusplus_api

identities) and testing sets (10 identities) have no overlap. AIST videos are public dataset only for the research purposes. The annotations of the AIST dataset is also public for research purposes and it is licensed by Google LLC CC-BY-4.0 license.