# OpenReview forum: "Neural Human Performer: Learning Generalizable Radiance Fields for Human Performance Rendering"
_NeurIPS.cc/2021/Conference — NeurIPS 2021 Spotlight_

### Official Review · Reviewer_5d4Y · 2021-07-04

**Rating:** 8
**Confidence:** 5

**Summary:**

The paper solves a problem similar to Neural Body (reconstruction / free-view rendering of clothed humans from videos based on SMPL fits) but does not require training a neural network per scene; instead, it generalises from training sequences and, at test time, runs only a feed-forward model conditioned on keyframes. This is important for many applications since training a neural net per scene is not practical, and also the proposed extension is not trivial: the paper brings novel ideas (using attention for time-wise and camera-wise aggregation). Experimental evaluation is comprehensive and the quality is even superior to per-scene optimised Neural Bodies. Given this, I suggest to accept the paper, although there is some room for improvement.

**Limitations And Societal Impact:**

In general, limitations section is good. I would suggest several improvements:
* please be clear upfront about the requirement to have SMPL fits as input; they are not given for granted, while Introduction does not mention this requirement. For other categories like animal species such parametric models are not available, so it limits the applicability of the method;
* the problem statement requires capture from static cameras only (extrinsics are not time-dependent): please make it clear and discuss whether it would be difficult to extend the method to support capture from e.g. one or several phones or AR devices;
* similar for prediction-time: the experiment al protocol seems to fix cameras: is it a big limitation? The supplementary videos seem to show free-view trajectories; could you comment how well it generalises? E.g. what if the cameras are sampled outside of a hemisphere manifold?

**Main Review:**

Novelty.

(+) The paper significantly improves over Neural Body [31] (which was formally published after NeurIPS submission deadline). To be able to generalise to varying scenes, it conditions on the images filmed by different cameras and/or at different timestamps. This idea is exploited in recent pre-prints but for generic object categories (PixelNerf, PVA, WCE); this paper for the first time 1) exploits parametric body model to make it applicable to an articulated category of humans, 2) proposes to use transformers for aggregation across time and cameras instead of simpler average pooling. This is novel even w.r.t. unpublished work.

Clarity.

(+) The paper is well written, it mostly provides enough details.

(−) The main miss is that it does not describe the procedure of fitting SMPL to videos, which is not easy to do in a time-consistent manner (is it reconciled across cameras and time?). The paper only refers to [7, 31, 9], which is not enough. Also it would be good to clarify early on (before Section 4) that the method uses estimated SMPL parameters as it affects applicability.

(−) The paper can be more straightforward about limitations, see the next section for details.

(−) Skeletal features are sampled from projected SMPL vertices. How does the method deal with occlusions? The paper also says the query skeletal features are obtained by trilinear sampling but does not provide more details. I understand that the function is defined on the SMPL template surface; how exactly are the features extrapolated for the outside volume?

(+) Related work is comprehensive.

Significance.

(+) Training a neural network per scene is impractical in many applications due to time or storage constraints. Making a generalisable model is a life saver in these cases.

(+) The method can be used both for free-view rendering and reconstruction of clothed humans. As shown, the competing methods produce wrong 3D even if the rendering looks reasonable. The output also looks temporarily smooth in videos, which is also a desirable property.

Quality.

(+) Experimental results are comprehensive. The paper ablates the contributions, including turning off pixel-aligned and skeleton-defined features separately, and it shows that both transformer architectures bring improvement over average pooling.

(=) In the NHR baseline, is it fair to use just SMPL vertices? Will the method improve significantly with higher resolution?

(=) AIST numbers (Table 2b) are significantly lower than ZJU ones; is it still meaningful to compare results with such low PSNR?

Notes to the authors (no effect on the rating):
* In the time-wise transformer, would it make sense to use positional encoding of time?
* line 165: the different geometry patterns ← different geometry patterns;
* line 184: Is the loss indeed just L2 norm (RMSE), or squared (MSE)?
* line 223: a health margin ← a healthy margin;
* lines 277, 279: “Table. 4”.
* in Tables 4 and 5, along with reconstructions, it would be interesting to show the SMPL fits, to see how far the methods deviate.

=====

UPD. I am satisfied with clarifications in the rebuttals, and raise the rating to 8 assuming they will be included in the paper.

**Time Spent Reviewing:**

4 hours

---

> ### Author Response · Authors · 2021-08-10
> **Reply to Reviewer 5d4Y**
>
> We appreciate the reviewers for the thoughtful feedback and are happy that the reviewers acknowledge the novelty of our method. Our answers to the questions are as follows.
>
> - - -
>
> Q1. SMPL fitting
>
> A1.
>
> * Is the SMPL fitting reconciled across cameras and time?
>
> The SMPL fitting is reconciled across cameras, but isn't reconciled across time (SMPL fitting is done in a frame-by-frame manner).
>
> * SMPL fitting procedure
>
> Thank you for letting us clarify this. In our work, we used the SMPL parameters provided by the ZJU-Mocap and AIST datasets. They use [b] and [c] for their SMPL estimation, respectively. In both ZJU-Mocap and AIST works, the SMPL parameters are estimated as follows. The SMPL estimation pipeline starts with frame-by-frame 2D keypoint detection and camera parameter estimation. The 2D keypoints are then triangulated to 3D keypoints. After that, bundle adjustment is performed to optimize the camera parameters and the 3D keypoints. Finally, the SMPL model is fitted with respect to the 3D keypoints in a frame-by-frame manner.
>
> [b] "EasyMocap. https://github.com/zju3dv/EasyMocap"
>
> [c] "AIST++ API.https://github.com/google/aistplusplus_api"
>
> * Clarify before section 4 that our method requires SMPL estimation
>
> We will also clarify in the Introduction and Method sections that our method uses estimated SMPL parameters.
>
> - - -
>
> Q2. Skeletal feature sampling
>
> A2.
>
> * How is the occlusion handled?
>
> We have experimented both (1) with and (2) without considering the occlusion. In case of (1), given the SMPL fits and the camera parameters, we compute the visible vertices by rasterization and we use image features only for these vertices. In case of (2), we compute 2d-projected image features for every vertex [Pixel-Nerf, PVA]. We empirically found that there was no significant difference between (1) and (2). We conjecture this is probably because our multi-view transformer implicitly learns to handle the occlusion. Therefore, in our final version, we use (2), i.e., compute 2d-projected image features for every vertex. We will clarify this in the final revision.
>
> * How are the features extrapolated for the outside volume? (Feature sampling for query points not lying on the SMPL surface)
>
> In our framework, query points are sampled in the 3D bounding box of the SMPL mesh (enlarged by 2.5% to cover the gap between the real geometry and the SMPL fit - please refer to the supplementary material B.1 Query point sampling details). Since the skeletal features, which are defined on the surface of the SMPL mesh, are relatively sparse in the 3D space, we first diffuse the skeletal features into a volume, which is the same size of the enlarged 3D bounding box of the SMPL fit, using the SparseConvNet [d]. For each query point inside this volume, we use trilinear sampling to compute its skeletal features. For the implementation details, please refer to the supplementary material B.1-Sampling of time-augmented skeletal feature w.r.t. a query point x.
>
> [d] "3D semantic segmentation with submanifold sparse convolutional networks"
>
> - - -
> Q3. NHR comparison
>
> A3. We agree that NHR performance would improve when tested on higher resolution point clouds obtained from dense viewpoints. NHR requires point clouds obtained from 50~80 cameras, while our framework is designed to work with sparse input viewpoints. Since it is difficult to obtain high quality point clouds from such sparse viewpoints, we provided the SMPL vertices instead as input to NHR. This is the same comparison setting introduced in the NeuralBody work. If required, we can provide more results in the final version by increasing the resolution of the points sampled on the SMPL mesh. However, we would like to point out that our focus is to generalize across multiple identities while NHR is a per-scene optimization method.
>
> - - -
> Q4. AIST numbers
>
> A4.
>
> * AIST scores are lower than ZJU-Mocap
>
> There are some reasons that the AIST scores are lower than the ZJU-Mocap numbers:
>
> (1) Input video resolution is low: Compared to the ZJU-Mocap dataset, the subject is farther away from the cameras, leading to lower resolution observation of the subject (our experiments use center-cropping around the very small foreground subject in the original dataset).
>
> (2) Since the dataset consists of complex dance sequences, the input videos have significant motion blur.
>
> (3) SMPL parameters derived from the input videos are relatively inaccurate as well due to (1) and (2).
>
> * Is it still meaningful to compare on AIST with low PSNR?
>
> Our method still provides reasonable quality novel view synthesis and 3D reconstruction results as seen in Figures 4, 5 and in the supplementary video. The tendency in these visual results of all compared methods is clearly consistent with their relative quantitative results (PSNR, SSIM), indicating the comparison is meaningful.
> Also, since the AIST dataset has more identities with more diverse actions, we believe that it is suitable to test the generalizability of our method.
>
> - - -
>
> Q5. Notes to the authors
>
> A5.
>
> * Corrections and suggestions
>
> Thank you for the constructive feedback. We will reflect your comments in the final version.
>
> * Would it make sense to use temporal positional encoding in the time-wise transformer?
>
> Thanks for the suggestion. It would be interesting to see the effectiveness of adding temporal positional encoding as the relative time distance between frames could indicate the importance of one frame to another. For example, a frame too far away from the current timestep may have different lighting conditions, and would ideally have low weight in the temporal aggregation.
>
> * Clarification on loss
>
> Yes, the loss used is MSE only.
>
> - - -
>
> Q6. Mention the requirement for SMPL fits
>
> A6. We will clarify in the introduction and methods part that our method requires SMPL fits as input.
>
> - - -
>
> Q7. The problem statement requires capture from static cameras only.
>
> A7. We would like to note that our algorithm does not have an explicit assumption of static cameras. The time-variable camera parameters are not expressed in the current problem statement because all sequences in the used datasets (ZJU-Mocap, AIST) are captured from static cameras.
> In practice, it might be hard to estimate the inputs to our method (SMPL fits, camera parameters, foreground masks) with moving cameras due to motion blur, changing background and lighting, and synchronization issues etc. We consider this as an orthogonal problem and expect that any advancements in unconstrained multi-view capture setups will help to generalize our method to in-the-wild settings. As long as the inputs above can be obtained accurately, our method can work on videos captured from moving cameras.
>
> - - -
>
> Q8. Inference time camera settings
>
> A8.
>
> * Does the system require static camera settings?
>
> As mentioned in Q7, our framework is not restricted to static camera settings.
>
> * How well it generalizes to new views (e.g., cameras sampled outside the hemisphere manifold)
>
> In our free-viewpoint rendering, we sample cameras from different distances and orientations (that are not constrained by the hemisphere manifold). Our method can generalize reasonably well to the free-view trajectories as can be seen in the supplementary video, since we have effectively constructed 3D representation from our skeletal features.

---

### Official Review · Reviewer_8dkm · 2021-07-09

**Rating:** 7
**Confidence:** 4

**Summary:**

This paper presents a generalizable NeRF network that can produce novel view synthesis for novel dynamic human scenes from sparse camera views, in a feed-forward manner. It leverages visual features from tracked parametric body model and similar to PixelNeRF, it conditions the NeRF representation on pixel-aligned image features for appearance generalization. A novel combination of a temporal Transformer and a multiview Transformer is proposed to integrate multi-frame and multi-view visual observations. It demonstrated stronger generalization capability than prior work.


**Limitations And Societal Impact:**

Well addressed.

**Main Review:**

Creating a free-viewpoint video synthesis using a general model is a highly challenging problem.  The paper learns a generalizable NeRF for various dynamic human scenes, with a major contribution on a novel combination of a temporal Transformer and a multiview Transformer that enables better integration of visual cues across time and views. Even though the components of leveraging geometric surface prior (SMPL as done in NeuralBody) and pixel-aligned image features (as in PixelNeRF) for human NeRF learning are not new, the work has achieved stronger generalization capability and modeling power for dynamic human scenes with the proposed learning framework.

The paper is well written and easy to follow. The experiments are extensive and well structured, demonstrating strong generalization capability to novel poses and identities compared with SOTA approaches.

There are a few places that could be improved.
- It is not clear how important the accuracy of the parametric body tracking is, during at both training and testing stage. The paper demonstrates the experiments on AIST and ZJUMocap, both of which are indoor scenes with 3d mocap data. Therefore I assume it is less a problem in obtaining accurate SMPL parameters estimation. However, for in-the-wild video datasets, the body tracking accuracy might largely degrade and I will be curious to understand how the proposed learning framework will perform.

- For the image evaluation metrics, it is not clear they are reported on novel camera views or the learned camera views.

- It will be good to see the cross-dataset generalization capability, even just between AIST and ZJUMocap datasets.

**Time Spent Reviewing:**

4.5

---

> ### Author Response · Authors · 2021-08-10
> **Reply to Reviewer 8dkm**
>
> We appreciate the reviewers for the thoughtful feedback and are happy that the reviewers acknowledge the novelty of our method. Our answers to the questions are as follows.
>
> - - -
>
> Q1. How important is the accuracy of the parametric body tracking?
>
> A1. It is true that the performance of our method will be affected as the SMPL parameter accuracy degrades. This is the reason why the scores on the AIST dataset, where the motions are complicated and thus relatively difficult to obtain accurate SMPL parameters, are lower compared to the scores experimented on the ZJU-Mocap dataset. We can discuss this in the limitations section. It would be an interesting direction to jointly refine the SMPL parameters within our framework using differentiable rendering for in-the-wild applications.
>
> - - -
>
> Q2. Are the image evaluation metrics are reported on novel camera views or learned camera views?
>
> A2. All evaluation metrics are reported on the novel camera views that are never seen at training.
>
> - - -
>
> Q3. It will be good to see the cross-dataset generalization capability.
>
> A3. Thanks for the suggestion. Please refer to Reviewer-r7DT Q4-A4, where we report cross-dataset generalization experiments.

---

### Official Review · Reviewer_ZRhs · 2021-07-15

**Rating:** 7
**Confidence:** 4

**Summary:**

This work proposes a method that can synthesize free-viewpoint videos for dynamic humans in a sparse multi-camera system.  As is claimed and demonstrated in the paper, by introducing transformer modules to aggregate spatio-temporal information in the sparse multi-view data, the learned neural radiance fields are generalizable to unseen human identities and unseen poses.
The proposed method is tested on public multi-camera datasets, showing promising results.

**Limitations And Societal Impact:**

Please see the main review.

**Main Review:**

The novelty of this work is good. Although the generalizability to unseen poses has also been achieved in a few concurrent works, the unseen identity generalizability makes the key difference to those per-scene optimization methods.

Some questions and comments,
1. it seems like that it is pretty straightforward to incorporate the Transformer for learning the generalizable radiance fields. So I am afraid that the technical contribution in this paper may not be very significant.

2. the exposition of the method section needs more effort to be improved.
     I would appreciate it much if the authors could make the method section much neater. The method section is full of tons of symbols, of which some may not be necessary, e.g., eq. 4 in Lin 178.
     Fig. 1 failed to do its job, I think the illustration of the method is supposed to be self-contained, it would be also much better to elaborate in the caption, instead of referring the readers back to each specific section.

3. regarding the details of the SMPL estimate. How many cameras are used for estimating the SMPL models of both ZJU-MoCap and AIST data? In Line 192, it is written that three methods are used to obtain these estimates, this confuses me. Also, the AIST dataset contains corresponding 3D keypoints and SMPL parameters (as described in the supplementary), are these data being used when running on AIST?

4. the paper did not show the failure cases. Is it true that the Transformers can always exploit the information in the memory bank to produce good imagery? What if the motions in the training data differ significantly from those in test data so that the memory bank fails. It is necessary to evaluate the transferability of the method in such a scenario, e.g., train on ZJU-MoCap, test on AIST, or vice-versa.

5. In Line 180, the samples are bounded within the bounding box of the SMPL mesh. But the SMPL mesh can be inaccurate, which means the bounding box derived from it can be inaccurate as well, e.g., the box may not be sufficiently large to enclose the GT shape.

Overall, this submission is good to me, and I hope the issues could be addressed in the rebuttal to make it a more solid submission.

**Time Spent Reviewing:**

Around 36 hours over several days

---

> ### Author Response · Authors · 2021-08-10
> **Reply to Reviewer ZRhs**
>
> We appreciate the reviewers for the thoughtful feedback and are happy that the reviewers acknowledge the novelty of our method. Our answers to the questions are as follows.
> - - -
>
> Q1. It seems straightforward to incorporate the Transformer for learning the generalizable radiance fields.
>
> A1. We would like to respectfully point out that we are the first to propose a combination of a temporal transformer and a multi-view transformer to integrate visual cues across time and cameras to be able to generalize across varying human identities.
>
> - - -
>
> Q2. The exposition of the method section needs to be neater.
>
> A2. We will address this in the revision.
>
> - - -
>
> Q3. Details of the SMPL parameters estimation.
>
> A3.
>
> * Which methods are used to estimate SMPL parameters?
>
> Sorry for the confusion. In our work, we used the SMPL parameters provided by the ZJU-Mocap and AIST datasets. They use [b] and [c] for their SMPL estimation, respectively. In both ZJU-Mocap and AIST works, the SMPL estimation pipeline starts with frame-by-frame 2D keypoint detection and camera parameter estimation. The 2D keypoints are then triangulated to 3D keypoints. After that, bundle adjustment is performed to optimize the camera parameters and the 3D keypoints. Finally, the SMPL model is fit with respect to the 3D keypoints in a frame-by-frame manner.
>
> [b] "EasyMocap. https://github.com/zju3dv/EasyMocap"
>
> [c] "AIST++ API.https://github.com/google/aistplusplus_api"
>
> * Are the 3D keypoints required to run Neural Human Performer on AIST?
>
> Our model does not require any 3D keypoint inputs regardless of dataset.
>
> - - -
>
> Q4. Failure cases
>
> A4.
>
> * What if the motions in the training data differ significantly from those in test data so that the memory bank fails?
>
> As we show in the paper and the supplementary video, our method can generalize well to unseen poses. We can visualize the distribution of the body poses observed both in training and testing motion sequences via t-SNE embedding in the final version to better demonstrate the pose variation we can handle.
> As with many deep learning approaches, we would like to note that as the distribution of testing motions becomes significantly different than the training data, our framework might not capture motion dependent details while the coarse overall geometry is still well represented thanks to the skeletal features learned on the SMPL surface. We will discuss this in the final revision.
>
>
> * Cross-data transferability
>
> Thanks for the suggestion. Please refer to Reviewer-r7DT Q4-A4, where we report cross-dataset generalization experiments.
>
>  - - -
>
> Q5. Sometimes inaccurate 3D bounding boxes may not enclose the GT shape.
>
> A5. In practice, we empirically found that using slightly enlarged bounding boxes (+2.5%) can sufficiently address the gap between the SMPL and the real geometry. We use this enlarged bounding box when sampling points during volume rendering. The details are also mentioned in the supplementary material (section B.1 Implementation details - Query point sampling details)

---

### Official Review · Reviewer_r7DT · 2021-07-17

**Rating:** 8
**Confidence:** 4

**Summary:**

This paper presents a method to synthesize a novel view image of a person given the sparse multiview images. The paper proposes time-augmented skeletal features---the pixel-aligned features that encode NeRF-like rendering fields by aggregating image features across  time and views, parametrized by a human mesh model. They use transformer networks to learn fusing multiview image features over time. The resulting features are used to generate an implicit function that can predict RGB and opacity. They demonstrate that the proposed method is generalizable to unseen identities and poses, validated on ZJU-MoCap and AIST datasets.


**Limitations And Societal Impact:**

Yes.

**Main Review:**

Strength

(+Novelty) This paper is built upon Neural Body [31] framework (NeRF+SMPL) that parametrizes the neural radiant field using the SMPL vertices, which synthesizes a novel view image given a set of multiview images. This paper takes a new step to learn a visual representation of humans using pixel-aligned features that can be used to generalize to new subjects. In this sense, it combines NB and PIFu, which is novel and shown to be very effective. The image feature is in particular useful for occluded regions of body because the learned image features are rather viewpoint invariant.  Further, they propose a novel way to fuse multiview and time features through self-guided attention through transformers, which shows stronger performance compared to existing average or max-pooling (similar idea for multiview fusion has been used in [a] though).

[a] Stable view synthesis, CVPR 2020

(+Performance) The experiments with multiple baselines (PixelNeRF, NB) show that the proposed image feature learning is highly effective on generalization of subjects and poses in particular, for unseen subjects and poses in terms of PSNR and SSIM metrics. The result in the supplementary video well illustrates the strengths of the proposed method. Further, the paper includes substantial ablation study that shows the impact of each component.


Weakness:

(Justification of transformer) Learning feature fusion over views is interesting but not sure how effective it is (do we really need to learn, or can use a weighted average based on viewing directions to fuse similar to [a]?). I suspect that performance wise, it will be pretty similar. If it is not way better than such weighted average skeme, having transformer over views is overkill and may limit the generalizability to a new camera configuration.

(Justification of temporal aggregation) Table 1-c illustrates the performance upper bound of the proposed method, which is on par with NB. This indicates temporal aggregation is not that effective. I thought the main claim of using temporal aggregation is that it can observe occluded regions as the performer changes poses so the features on the occluded regions can be taken into account. However, this is not reflected in the Table. I would suggest to conduct experiments that can justify the temporal aggregation for such a controlled experiment (this is different from the one in the ablation study).,



More comments:
-R^{3->2} to R^{3}->R^2
-L141: what does it mean by second axis?
-Since the network is supposed to learn human appearance agnostic to camera poses, it would be nice to demonstrate cross-dataset evaluation, e.g., learning from ZJU and testing on AIST.
-Adding the qualitative results for Table 1-(c) would be useful to see the performance upper bound of the proposed rendering method.



**Time Spent Reviewing:**

6

---

> ### Author Response · Authors · 2021-08-10
> **Reply to Reviewer r7DT**
>
> We appreciate the reviewers for the thoughtful feedback and are happy that the reviewers acknowledge the novelty of our method. Our answers to the questions are as follows.
> - - -
>
> Q1: Justification of multi-view transformer.
>
> A1:
> * Performance aspect:
> To validate the effect of the multi-view (MV) transformer, we conducted an ablation over various view aggregation methods: 1) plain averaging [Pixel-Nerf], 2) view-dependent weighted averaging [a: Stable view synthesis], 3) camera-encoded learned averaging [PVA] and 4) our proposed MV transformer. We use our model without the temporal transformer (5th-row of Table 3) in this experiment to better investigate the effect of each averaging method. Following table confirms the effectiveness of the MV transformer with healthy gains (>0.8 PSNR, >1.2% SSIM) above all other methods. Also, other aggregation methods (1, 2, 3) often show artifacts similar to those in Figure 3. We can add quantitative and qualitative results in the final version.
> |       Method        | PSNR  | SSIM   |
> |------------------------|-------|--------|
> | 1. Plain avg              | 23.47 | 0.8906 |
> | 2. View-dependent avg    | 23.60 | 0.8912 |
> | 3. Camera-encoded avg     | 23.39 | 0.8715 |
> | 4. Multi-view transformer | 24.44 | 0.9034 |
>
> * Generalization aspect:
> We show that our model can effectively generalize to novel camera configurations in the supplementary video: free-viewpoint rendering results.
> - - -
>
> Q2. Justification of temporal aggregation.
>
> A2. We would like to first point out that the comparison in Table 1-c places our method in disadvantage. This is because our model is trained on all the source subjects at once (one network for 7 subjects), while the competing methods (e.g., NB, NHR, NT) are per-subject trained (7 networks, one for each of the 7 subjects) - easier setting. Even so, we obtain on-par results with NB.
> To make a more fair comparison, we conduct an additional experiment in the exact same per-scene training setting. The following table shows our model outperforms NB by +3 PSNR and +1.4% SSIM. We also confirm the effectiveness of the temporal transformer, where removing it leads to notable drops of -0.9 PSNR and -1% SSIM.
>
> |        Exp. protocol           |   Method             | PSNR  | SSIM    |
> |---------------------------------|-------------------------|-------|---------|
> | Per-subject trained        | | |
> |                                          | Neural Body             | 28.51 | 0.947   |
> |                                         | Ours w/o T-transformer  | 30.72 | 0.957   |
> |                                         | Ours with T-transformer | **31.57** | **0.966**   |
> | Trained on all subjects at once | | |
> |   | Ours with T-transformer | 28.73 | 0.936   |
>
> - - -
>
> Q3. Line 141 - what does it mean by 'second axis'?
>
> A3. The second axis is the time axis over the memory frames. We apply softmax across the memory frames to compute the soft weights of different timesteps w.r.t. the current timestep. For more details, please refer to the supplementary: B.1 line 16-19 and Figure 1.
>
> - - -
>
> Q4. Cross-dataset generalization capability
>
> A4. We report the cross-dataset generalization scores on unseen subjects and poses (AIST -> ZJU-Mocap / ZJU-Mocap -> AIST generalizations).
>
> We would like to first point out that the two datasets have highly different color distribution (background, lighting) and distance of camera to the subject, which makes the cross-dataset generalization extremely challenging. Nevertheless, we visually observed that our method can still capture the overall geometry thanks to the skeletal features. Our model outperforms all the competitors in this setting (see below). Furthermore, we found that only 5-minute fine-tuning on the target dataset can already outperform the baselines fully-trained on the target dataset, and 10-minute fine-tuning performs on par with our model fully-trained on the target dataset.
>
> * Generalization tested on ZJU-Mocap
> |          Exp. protocol                 | Method  | PSNR  | SSIM   |
> |------------------------------------------------|------------|-------|--------|
> | Trained on ZJU-Mocap                     |            |       |        |
> |                                                | Pixel-Nerf | 23.17 | 0.8693 |
> |                                                | PVA        | 23.15 | 0.8663 |
> |                                                | Ours       | 24.75 | 0.9058 |
> | Trained on AIST (no fine-tune)                 |            |       |        |
> |                                                | Pixel-Nerf | 12.32 | 0.5948 |
> |                                                | PVA        | 12.28 | 0.5801 |
> |                                                | Ours       | 17.05 | 0.7711 |
> | Trained on AIST (5-min fine-tune on ZJU-Mocap) |            |       |        |
> |                                                | Ours       | 24.24 | 0.8930 |
> | Trained on AIST (10-min fine-tune on ZJU-Mocap)  |            |       |        |
> |                                                | Ours       | 24.71 | 0.9013 |
>
> * Generalization tested on AIST
> |                    Exp. protocol                 |  Method  | PSNR  | SSIM   |
> |------------------------------------------------|------------|-------|--------|
> | Trained on AIST                     |            |       |        |
> |                                                | Pixel-Nerf | 18.06 | 0.7304 |
> |                                                | PVA        | 17.82 | 0.7211 |
> |                                                | Ours       | 19.03 | 0.8390 |
> | Trained on ZJU-Mocap (no fine-tune)               |            |       |        |
> |                                                | Pixel-Nerf | 11.34 | 0.5402 |
> |                                                | PVA        | 11.29 | 0.5477 |
> |                                                | Ours       | 15.33 | 0.7545 |
> | Trained on ZJU-Mocap (5-min fine-tune on AIST) |            |       |        |
> |                                                | Ours       |18.62 | 0.8238 |
> | Trained on ZJU-Mocap (10-min fine-tune on AIST)  |            |       |        |
> |                                                | Ours       | 18.80 | 0.8352 |
>
> - - -
>
> Q5. Add the qualitative results for Table 1-c.
>
> A5. Thanks for the suggestion. We will include it in the final version.

---

### Decision · Program_Chairs · 2021-09-27

**Decision:**

Accept (Spotlight)

**Comment:**

This submission received 4 positive final ratings: 8, 7, 7, 8.
The reviewers appreciated overall novelty of the approach, its generalization properties and strong empirical performance. The remaining concerns were mostly around clarity, justification of individual components and lack of analysis of failure cases. These were mostly addressed in the rebuttal, as acknowledged by the reviewers. The final recommendation is therefore to accept as a spotlight.